# A meta-learning approach for selectivity prediction in asymmetric catalysis

Sukriti Singh [ORCID] ✉ & José Miguel Hernández-Lobato [ORCID] ✉

Transition metal-catalyzed asymmetric reactions are of high contemporary importance in organic synthesis. Recently, machine learning (ML) has shown promise in accelerating the development of newer catalytic protocols. However, the need for large amount of experimental data can present a bottleneck for implementing ML models. Here, we propose a meta-learning workflow that can harness the literature-derived data to extract shared reaction features and requires only a few examples to predict the outcome of new reactions. Prototypical networks are used as a meta-learning method to predict the enantioselectivity of asymmetric hydrogenation of olefins. This meta-learning model consistently provides significant performance improvement over other popular ML methods such as random forests and graph neural networks. The performance of our meta-model is analyzed with varying sizes of training examples to demonstrate its utility even with limited data. A good model performance on an out-of-sample test set further indicates the general applicability of our approach. We believe this work will provide a leap forward in identifying promising reactions in the early phases of reaction development when minimal data is available.

The burgeoning demand for chiral compounds in applications ranging from pharmaceutical to material science has greatly advanced the field of asymmetric catalysis. The transition metal-catalyzed asymmetric hydrogenation of olefins (AHO) is among the most important synthetic methods to obtain high-purity chiral molecules[1,2]. The interest in AHO is primarily due to its broad scope and efficiency, including high selectivity and atom economy[3]. This has led to significant developments in the design of new metal catalysts and chiral ligands, thereby increasing the reaction scope to hydrogenate a variety of substrates with high enantioselectivity[4]. Despite the progress, there has been challenges in the asymmetric hydrogenation of unfunctionalized and/ or tetrasubstituted olefins[5]. The traditional approaches to the design of asymmetric catalysts are usually time and resource intensive. Therefore, the development of new methods utilizing the data-driven approaches to streamline the tedious process of reaction development is of high interest.

In recent years, there has been a rapid growth in employing machine learning (ML) methods for the exploration of high-dimensional chemical space[6,7]. Some examples include the use of supervised ML algorithms to predict molecular properties (e.g., toxicity, solubility) as well as reaction outcomes such as yield, regioselectivity, stereoselectivity, and so on[8-11]. Unsupervised ML techniques like dimensionality reduction and clustering are frequently utilized for the visualization of reactivity patterns to guide catalyst design and select substrate scope[12,13]. Lately, reaction optimization to identify optimal conditions using Bayesian optimization (BO) has also gained interest[14,15].

The potential of an ML model to provide useful predictions is crucially dependent on the training data[16,17]. This data is usually derived from the published literature or high-throughput experimentation (HTE). Although the datasets obtained from HTE are of high-quality and well-standardized, they are publicly available only for a few popular reaction types[18]. On the other hand, the published literature has limitations regarding the quality of data but is present in large amounts, covering a broad range of reaction classes. Owing to the challenges in generating data from HTE, there is a need to develop methods that can use the literature data and provide good predictive accuracy[19]. The literature-derived datasets have a high variation in

Department of Engineering, University of Cambridge, Cambridge, UK. ✉e-mail: sukriti243@gmail.com; jmh233@cam.ac.uk

individual reaction components, but the diversity in terms of reactant-product combination is generally low[20]. Thus, an ML model is required that can capture knowledge of related reactions from the literature data and provide good predictions with few examples. Meta-learning has recently emerged as the popular learning strategy that teaches models how to learn efficiently under these circumstances[21].

Meta-learning involves training a meta-model on multiple related tasks with the goal of extracting shared knowledge and leverage this information for quick adaptation to an unseen task with limited data[22,23]. The computer vision domain has been the primary focus of the advancements in meta-learning, in particular, few-shot learning to address low-data problems[24,25]. In the past few years, few-shot learning has also shown some promise in drug discovery for enhancing molecular property prediction and optimization in low-data regimes[26–31]. Additionally, a few-shot yield prediction model on a HTE dataset is also reported[32].

The ML models have been extremely useful for reaction outcome prediction on HTE datasets[33,34]. However, predicting reaction outcomes using a broad selection of published reaction data remains challenging. This is due to the sparsity in literature datasets where only a few reactions are experimentally known out of an immeasurably large accessible chemical space[35,36]. To fill this gap, we design a meta-learning model that can significantly improve predictive performance on datasets derived from the literature. We demonstrate our protocol on a literature-mined dataset focused on transition metal-catalyzed asymmetric hydrogenation of olefins. We also show the applicability of our meta-model to generalize and make predictions in low-data settings. The proposed approach can be used to train a meta-model for any reaction of interest. This would be beneficial during the initial stages of reaction development to quickly predict the outcome for new samples in a few-shot manner.

## Results and discussion

In a recent study, the Hong group reported a database containing over 12,000 literature-derived reactions for transition metal-catalyzed asymmetric hydrogenation of olefins[37,38]. For each reaction, the dataset consists of information on substrate, metal, ligand, product, solvent, additive, pressure, temperature, and catalyst loading (Fig. 1). The reaction performance is included in terms of enantioselectivity, yield, and turnover number (if available). Among the reported reactions, 90% are catalyzed by rhodium and iridium metal catalysts. Whereas, cobalt catalysts account for a mere 1% of the reactions. For cobalt-catalyzed AHO reactions, we extracted additional data from published literature in the last few years. Herein, we only consider the AHO reactions catalyzed by iridium, rhodium, and cobalt metal catalysts which are associated with the same group in the periodic table. The dataset contains a total of 11,932 AHO reactions of which 5009 belongs to Ir, while 6391 and 532 reactions are catalyzed respectively by Rh and Co. In our recent work, a detailed data-driven analysis of this AHO dataset is carried out[39]. With the use of informative plots and figures, the type of olefins and ligands used in the AHO reactions catalyzed by Ir, Rh, and Co metal catalysts are described. The trends in reactions conditions, for example, solvents, temperature, and pressure are also discussed.

### Feature representation

To encode the reaction space, two widely used feature representations, that is, Morgan fingerprints[40] and molecular graphs are considered. In the first case, Morgan fingerprints for olefin, ligand, and solvent are computed as a 512-bit vectors with radius 2. The identity of metal and presence/absence of the additive is incorporated in the representation using one-hot encoding. The reaction condition comprising of pressure, temperature, and catalyst loading is included as well. The composite reaction representation is a concatenation of fingerprints and one-hot encodings of individual reaction components along with the reaction condition (Fig. 2a). This results in an input representation with a dimension of 1544. As an alternative feature representation, the molecular graphs of olefin, ligand, and solvent are fed into graph neural networks (GNNs) to extract feature embeddings. For each reaction component, the graph with atom and bond features is passed to the message-passing neural network[41]. Three message-passing steps are considered to obtain the node representation, where an edge network is used as the message function and a gated recurrent unit (GRU) as the update function. The node representation vector has a dimension of 64. In the readout step, a set2set model[42] (with number of set2set layers fixed to 3) provides global pooling over the node representation vectors. This results into a graph representation vector of dimension 512. In this case, the full reaction representation is a concatenation of the graph embeddings with one-hot encoded vectors, and reaction conditions (Fig. 2b). The reaction representation vector thus obtained has a dimension of 1544.

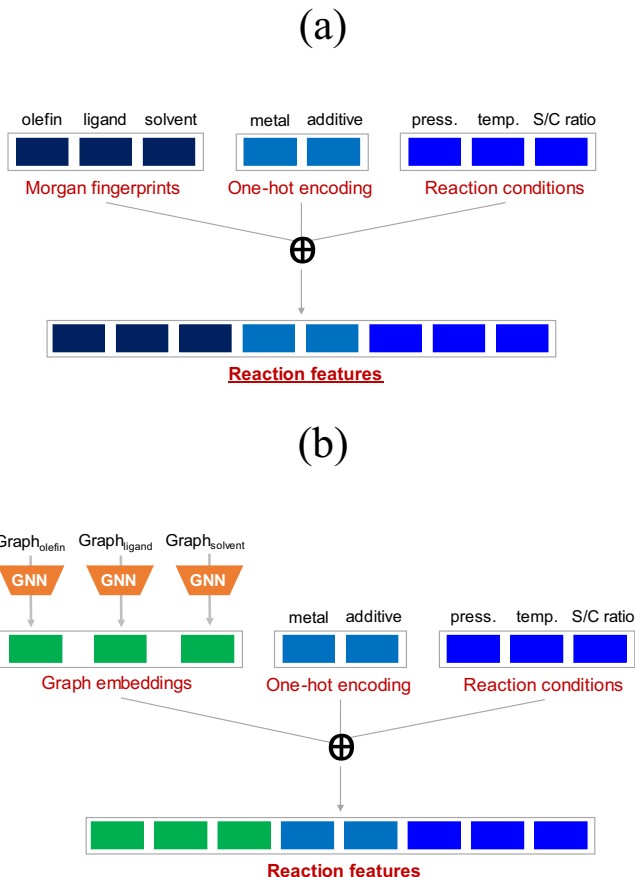

**Fig. 2 | Reaction representation.** A general description for obtaining a reaction feature representation from (**a**) Morgan fingerprints, and (**b**) molecular graphs. The S/C ratio corresponds to catalyst loading. The ⊕ symbol indicates concatenation of vector entries.

**Fig. 1 | Asymmetric hydrogenation of olefins.** Representative example showing the various reaction components involved in the asymmetric hydrogenation of olefins.

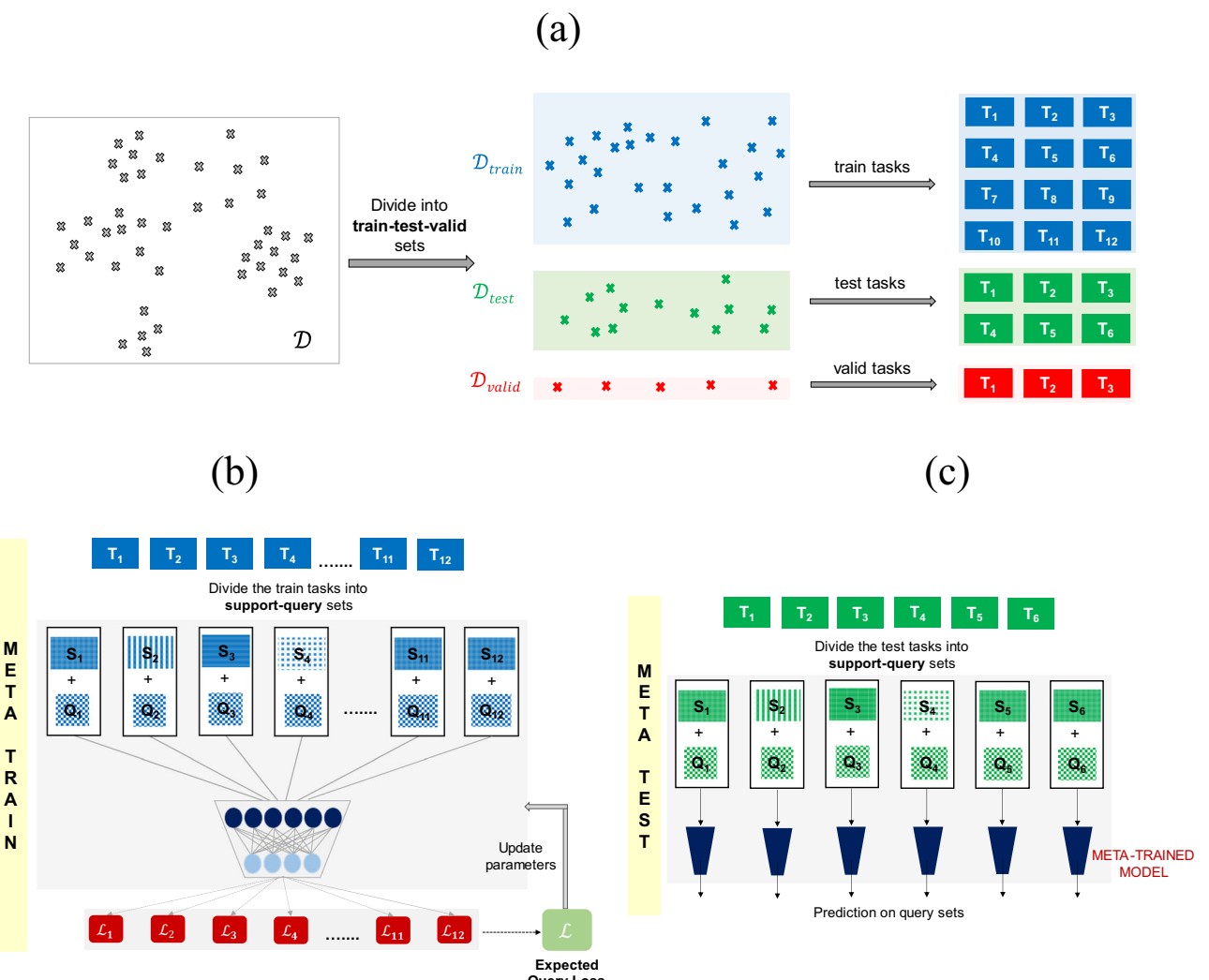

**Fig. 3 | Meta-learning workflow for task construction and model training.**
**a** Illustration of the steps involved in constructing tasks for meta-learning. The data $\mathscr{D}$ is first randomly split into train ($\mathscr{D}_{train}$), test ($\mathscr{D}_{test}$), and validation ($\mathscr{D}_{valid}$) set. Each set is then randomly partitioned into training, test, and validation tasks. A schematic representation of the process followed at (**b**) meta-train, and (**c**) meta-test time. During meta-training time, the model parameters are updated by optimizing the expected query loss over multiple training tasks. At meta-test time, the optimized parameters are used to obtain predictions on the query set, given the support set for the test tasks.

## Dataset preparation

Meta-learning has certain design requirements for the data[43]. We need a large set of training tasks for pretraining and a set of test tasks. The primary question, then, is how to construct these tasks. For this purpose, the reaction data $\mathscr{D}$ is first randomly partitioned into training ($\mathscr{D}_{train}$) and test ($\mathscr{D}_{test}$) sets, with 9032 and 2400 reactions respectively (Fig. 3a). A small amount of data with 500 reactions is kept for validation ($\mathscr{D}_{valid}$) as well. Now, the simplest approach to get tasks for training is to randomly partition $\mathscr{D}_{train}$ into subsets of data, with each subset representing a different training task. A similar approach is followed to obtain test and validation tasks (Fig. 3a). Once these tasks are constructed, a meta-learning model can be trained as follows. During the meta-training phase, the training tasks are further divided into support and query sets. The model is first trained on the support set. The model parameters are then learned to obtain the best possible average validation loss (on the query set) over multiple training tasks (Fig. 3b). During meta-testing, the test tasks are further split into support and query sets. The optimized parameters of the meta-trained model are then used to obtain predictions on the query set, given the corresponding support set (Fig. 3c).

## Model training and evaluation

In this section, we apply the meta-learning method in a binary classification setting. For this purpose, the reactions are split into two classes: (a) enantioselectivity (%ee) greater than 80, and (b) %ee lower than 80. A threshold value of 80 is decided based on the data distribution and a generally acceptable number for highly selective reactions. The resulting dataset has 65% of the reactions with %ee greater than 80, while only 35 % reactions have %ee lower than 80. The classification performance is evaluated using the area under the precision-recall curve (AUPRC), which is sensitive to class imbalance. In addition, the model performance in terms of the area under the receiver operating characteristic curve (AUROC) is also reported.

As discussed previously, the meta-training phase starts with partitioning each training task into a support and a query set (Fig. 3b). We sample a total of 512 reactions for the support set and 64 for the query set. The hyperparameters are tuned by evaluating the meta-model on the validation tasks with a support and query set of size 64 and 128 respectively (for implementation details see section 1 of the Supplementary Information). The meta-learner is then trained by iteratively feeding it the support sets of a batch of 5 training tasks and then

maximizing its average prediction log-likelihood on the corresponding query sets for those training tasks. At meta-test time, an average performance over ten different support and query random splits of every test task is reported. This process is performed for three support set sizes 16, 32, and 64 to analyze how the model performance varies with the amount of training data available in a new task. In each case, the query set size is fixed to 128 reactions. To demonstrate the utility of meta-training, we compare the model performance with several single-task methods. Random forests (RF) are known to work well in small data situations[44]. Whereas, GNNs generally require relatively larger datasets[45]. These single-task methods are trained from scratch using the support set of the test tasks, without leveraging the information of the meta-trained model.

The overall test performance of the meta-learning and single-task methods on the AHO dataset is presented in Fig. 4a (section 2 in the Supplementary Information). All the results are reported using fingerprints as the molecular representation (Fig. 2a). It is apparent from Fig. 4a that the prototypical network (Protonet) significantly outperforms single-task methods (RF and GNN). Protonet is able to achieve good performance gains even when a limited training data is available, in contrast to RF and GNN that perform poorly on these tasks. An increase in model performance is observed with the availability of more data in the support set of test tasks. To obtain a better comparison, we also computed the performance of single-task methods

trained on the full training set ($\mathscr{D}_{train}$). It can be noted from Fig. 4a that the performance of RF and GNN with full training data is comparable to Protonet with a support set size of only 32 examples. Additionally, Protonet with a support set size of 64 performs better than both single-task methods trained on all the training data $\mathscr{D}_{train}$. These results guided us to use $\mathscr{D}_{train}$ as the support set of the test tasks at meta-test time (Fig. 4b), instead of sampling the support set from the test task itself (Fig. 3c). Here, the original support set of the test tasks is not included in $\mathscr{D}_{train}$. Protonet with $\mathscr{D}_{train}$ as support set of the test tasks provided the best model performance with an AUPRC score of $0.9133 \pm 0.0031$. Whereas, RF and GNN yielded an AUPRC score of $0.8369 \pm 0.0055$ and $0.8259 \pm 0.0021$ respectively (Fig. 4a). The model performance of Protonet with molecular graphs as input representation is found to be comparable to that with fingerprints (section 3 in the Supplementary Information).

### An alternative approach for task construction in meta-learning

For any meta-learning method, the tasks are chosen in a manner to enable efficient learning. It generally requires diverse tasks, but also some similarity between a task's support and query set. This similarity is particularly important to obtain good predictive performance on the query set, with only a few examples in the support set. Given the diversity in AHO reaction data in terms of olefins, ligands, reaction conditions, and so on, it might be difficult to capture the similarity between support-query sets by the random splitting of data (Fig. 3a). In this regard, we decided to use unsupervised clustering to create our train and test tasks. The process of selecting the tasks is described in Fig. 5a. The first step involves clustering of the full dataset $\mathscr{D}$. To implement this step, the 1544-dimensional fingerprint representation of the reaction is employed (Fig. 2a). Given the challenges involved in clustering the high-dimensional data, the dimensionality reduction of the reaction space serves as an important preprocessing step to achieve better clustering performance[46,47]. The uniform manifold approximation and projection (UMAP) is used to reduce the dimension of the reaction space to $10$[48]. This is followed by the k-means algorithm to divide the dataset into distinct clusters (section 4 in the Supplementary Information). For each cluster, the data belonging to $\mathscr{D}_{train}$ and $\mathscr{D}_{test}$ constitutes train and test tasks respectively (Fig. 5a). A total of 10 clusters resulted into 10 train tasks and 10 test tasks.

Following this clustering approach, the meta-training procedure remains the same, as described in Fig. 3b. The difference appears only in the construction of the support set during meta-test time (Figs. 3c and 5b). In the standard meta-learning framework, support and query sets are random splits of the test task (Fig. 3c). In our alternative approach (denoted as meta-cluster), the query set is randomly sampled from the test task, but the support set is now part of the training task belonging to the same cluster as that of the test task (Fig. 5b). For example, at meta-test time, if the query set is sampled from the test task of cluster C1, then the corresponding support set is selected from the train task of C1. Similar to the previous model evaluation procedure, we use three support set sizes of 16, 32, and 64. In addition to sampling different support set sizes, we also utilize all the data present in the training task (for example, $T_{C1}$ in Fig. 5a) of any cluster (denoted as $\mathscr{D}_{cluster}$) as the support set during meta-test time. The query set size remains the same with 128 reactions.

The summary of results for the meta-cluster and single-task-cluster methods is shown in Fig. 5c (section 5 in the Supplementary Information). In addition to RF and GNN, the model performance of other single-task methods such as support vector machines and gradient boosting is also reported (Table S10 in the Supplementary Information). Similar to Fig. 4a, Protonet_cluster significantly outperforms single-task methods (RF_cluster and GNN_cluster) for all support set sizes. Also, the model performance improves with the increase in support set size. Next, we compared these results to that of the

(a)

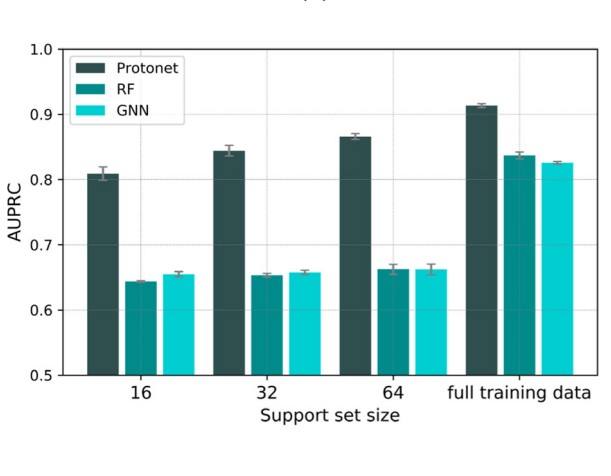

(b)

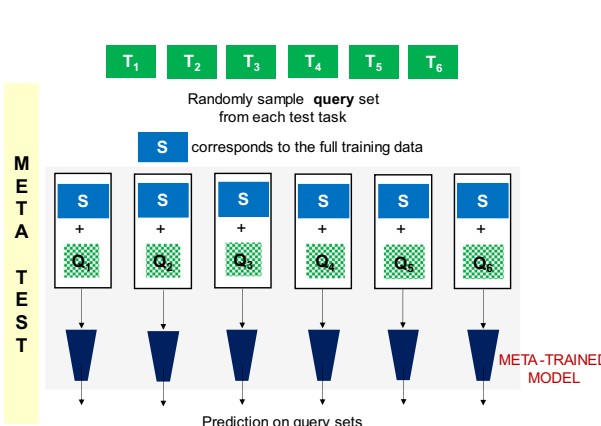

**Fig. 4 | Performance summary of meta-learning and single-task methods.**
a Summary of results for the meta-learning and single-task methods. b A schematic representation of the process followed at meta-test time with the full training data $\mathscr{D}_{train}$ used as the support set for test tasks. $T_1, T_2, ...., T_6$ (in green) represent test tasks as shown in Fig. 3a.

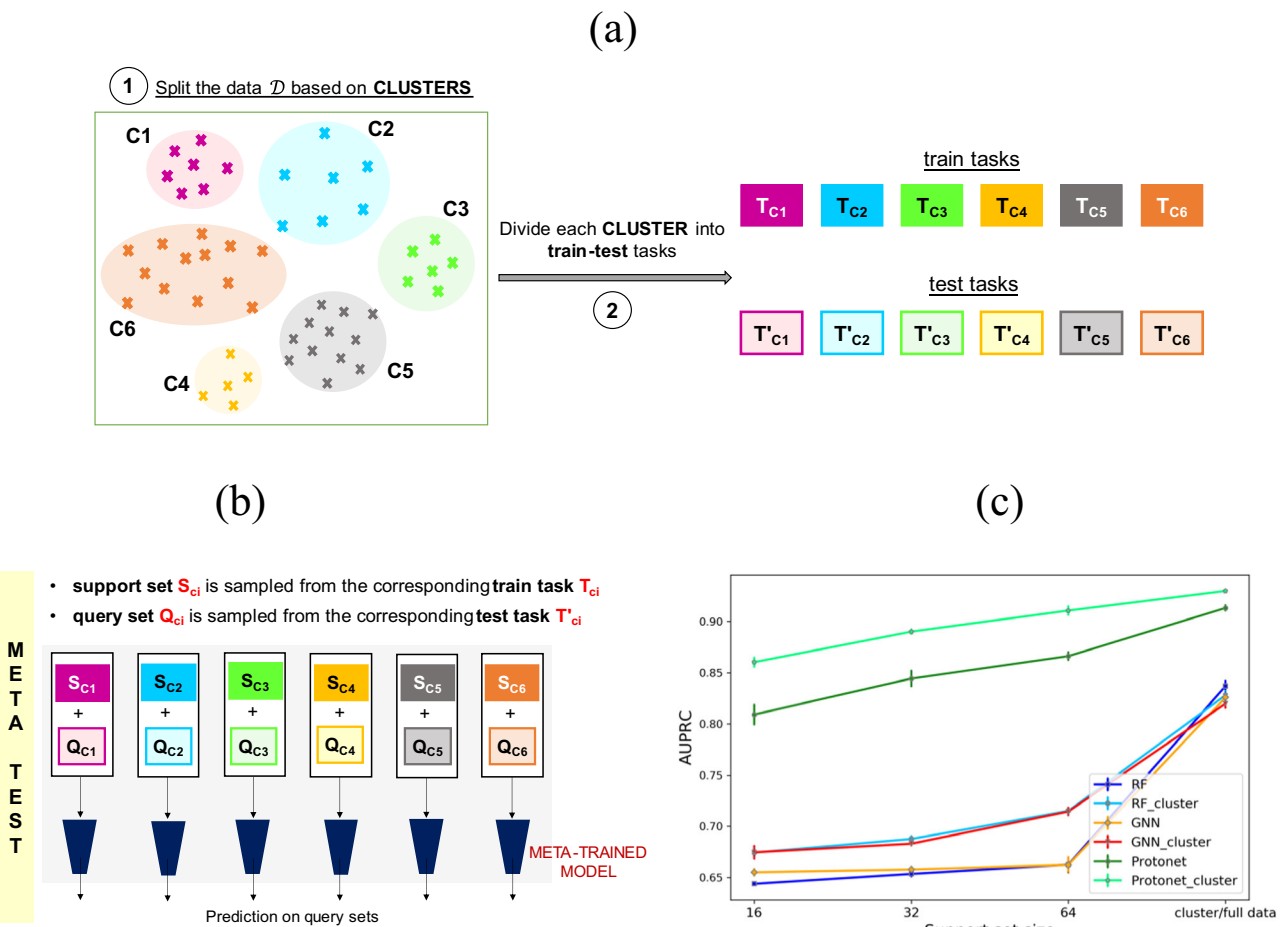

**Fig. 5 | Clustering-based meta-learning workflow for task construction and model evaluation. a** Illustration of steps involved in constructing train and test tasks for meta-learning based on clustering. The data $\mathcal{D}$ is first partitioned into various clusters using nonlinear dimensionality reduction followed by k-means clustering. Each cluster is then split into training and test tasks. **b** A schematic representation of the process followed to obtain support-query sets at meta-test time. **c** A comparative plot to analyze the effect of random and clustered support set for test tasks on model performance for both meta-learning and single-task methods. Cluster/full data on the x-axis denotes utilizing $\mathcal{D}_{cluster}$ and $\mathcal{D}_{train}$ respectively as the support set during meta-test time. For example, the results for RF_cluster and RF are with $\mathcal{D}_{cluster}$ and $\mathcal{D}_{train}$ as the support set, respectively.

standard meta-learning approach as discussed in the previous section (Fig. 4a). This approach is denoted as Protonet in Fig. 5c. It can be noticed from Fig. 5c that the clustering approach to construct the support sets of test tasks consistently results in better predictive performance for both meta-learning and single-task methods. Furthermore, Protonet_cluster (AUPRC = 0.9300 ± 0.0011) using $\mathcal{D}_{cluster}$ as support set performs better than Protonet that utilizes the full training data $\mathcal{D}_{train}$ in the support set (Fig. 5c). On the other hand, no significant difference is observed in the performance of RF_cluster (AUPRC = 0.8281 ± 0.0048) and RF with $\mathcal{D}_{cluster}$ or $\mathcal{D}_{train}$ as support set, respectively. A similar trend is observed for GNN_cluster (AUPRC = 0.8199 ± 0.0047) and GNN. Overall, the test performance appears to get benefited from a more similar support set during meta-test time. The gain is even more pronounced when only a small amount of data is present in the support set of the test tasks. After obtaining encouraging results from the meta-cluster approach during meta-test time, we decided to evaluate the impact of clustering on meta-training. Instead of randomly sampling the training tasks from $\mathcal{D}_{train}$ (Fig. 3a), each cluster is considered as a separate training task (Fig. 5a). A meta-model is then trained using these tasks and performance is evaluated on the test tasks, as shown in Fig. 5b. The meta-cluster approach with an AUPRC score of 0.9117 ± 0.0026, provided no significant gains during meta-train time (the process of meta-training is illustrated in Fig. 3b).

Moreover, we have also evaluated the performance of meta-cluster approach with two additional classification thresholds of 70 and 90 % ee (section 6 in the Supplementary Information). The meta-learning model resulted in improved model performance as compared to single-task method. Next, we analyzed the predictions of the Protonet_cluster model in terms of metal catalyst, type of olefin and ligand. The predictions of one of the test runs is considered (section 7 in the Supplementary Information). No particular trend is observed in the type of reactions for which the model struggles in making predictions.

Given the performance improvement obtained with the meta-cluster approach, it becomes important to demonstrate that the results are robust to (a) the process of clustering and (b) the selected number of clusters (section 8 in the Supplementary Information). For the first part, we repeated 10 times the process of selecting the support sets for the test tasks using clustering (Fig. 5a). Protonet_cluster obtained an AUPRC score of 0.9264 ± 0.0018 averaged over these 10 runs. This is comparable to the AUPRC score of 0.9300 ± 0.0011 with a single run. For the second part, we selected the number of clusters to be 10, 15, and 20. Protonet_cluster provided an AUPRC score of 0.9223 ± 0.0024, 0.9264 ± 0.0018, and 0.9248 ± 0.0016, respectively, averaged over 10 clustering runs. These results indicate that the meta-cluster approach requires only a set of related reactions for the

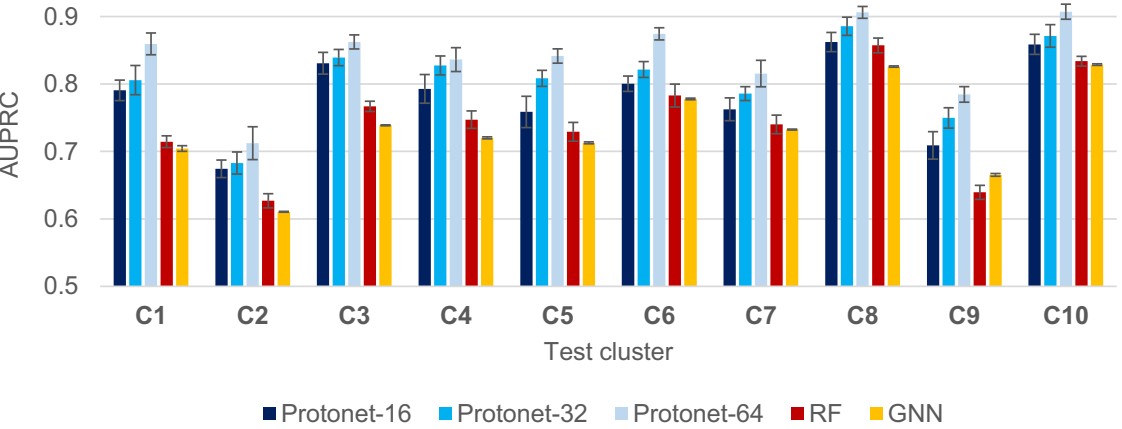

**Fig. 6 | Performance evaluation on cluster-based splits.** Comparison of performance of meta-learning and single-task methods on cluster-based splits. Protonet-16, Protonet-32, and Protonet-64 corresponds to prototypical networks trained on support set sizes of 16, 32, and 64 respectively. The performance of RF and GNN is reported with the model trained using full-training data. The error bars represent the standard error over 10 support-query splits.

support set of the test task and is robust to both the clustering process and the selected number of clusters.

Next, we evaluated the performance of the meta-learning method on more challenging cluster-based splits (section 9 in the Supplementary Information). A leave-one-cluster-out type approach is followed. Out of the 10 clusters obtained through clustering analysis, one cluster is kept as test task while remaining 9 clusters are used for training. This process is repeated for all 10 clusters such that each cluster becomes part of the test task at least once. The meta-testing procedure is same as shown in Fig. 3c, where support and query splits are part of the test task. The meta-model performance is evaluated on three support set sizes: 16, 32, and 64. The query set size remains 128 across all support sets. The result of prototypical networks on cluster-based splits is shown in Fig. 6. The performance is reported as an average over 10 random support-query splits of the test task. For comparison with meta-learning method, the performance of RF and GNN trained on full training data is also reported. It can be noted from Fig. 6 that prototypical networks return better predictions compared to single-task methods across all clusters. An increase in performance of meta-model is observed with the greater number of examples in the support set. In general, an AUPRC score of greater than 0.80 is observed across all clusters, except C2 and C9 where the model performance is relatively worse. On the other hand, C8 and C10 provide an AUPRC score of ~0.90, which is similar to that obtained with the meta-cluster approach where support and query are part of the same cluster (Fig. 5c).

## Model performance on out-of-sample test sets

The generalizability of our meta-model is further evaluated with an out-of-sample test set. To this end, we manually extracted another 245 Ir- and Rh-catalyzed AHO reactions from the recent literature, that are not a part of the original training data $\mathscr{D}$. Some of the substrates or ligands in the out-of-sample dataset are similar to that of the training data. But the overall reaction which is a combination of substrates, ligands, and reaction conditions (Fig. 2) is entirely different in out-of-sample test set and original training data $\mathscr{D}$. The reactions chosen are of high contemporary interest owing to their broad applicability. The diversity of the reactions involved in the out-of-sample test set can be appreciated from Fig. 7a. Reaction-1 and reaction-2 involve asymmetric hydrogenation of a varied range of di-, tri-, and tetrasubstituted non-chelating olefins[49,50]. Reaction-3 reports the Ir-catalyzed asymmetric hydrogenation of tetrasubstituted acyclic enones[51]. The tetra-substituted olefins are among the most challenging substrates and less-explored as compared to di- and trisubstituted olefins. Reaction-4

and reaction-5 presents the Rh-catalyzed asymmetric hydrogenation to obtain chiral 2-CF$_3$-chroman-4-ones derivatives and sulfones respectively[52,53]. Both of these are important structural moieties prevalent in many natural products and pharmaceuticals.

Now, the training set comprises of the full AHO reaction data with 11,932 reactions. The test set contains the out-of-sample reactions. We have employed both approaches (Figs. 3a and 5a) described previously to obtain the training and test tasks. In the first approach, the train and test tasks are constructed by randomly partitioning training and test data into subsets, with each subset representing a different task (Fig. 3a). The meta-model is trained on the full reaction data $\mathscr{D}$ using the same process as illustrated in Fig. 3b. Four different support set sizes of 8, 16, 32, and 64 are considered. The query set size is chosen to be 128. The model performance averaged over ten different support-query random splits of every test task is reported (section 10 in the Supplementary Information). It can be noted from Fig. 7b that Protonet performs significantly better than RF and GNN. A performance improvement with the increase in support set size is observed for all methods, but is more pronounced with Protonet. With a support set size of 64, Protonet obtains an AUPRC score of $0.9341 \pm 0.0053$, while it is $0.8631 \pm 0.0061$ and $0.8764 \pm 0.0090$, respectively for RF and GNN. It is to be noted that the support and query sets formed from the out-of-sample test data are already quite similar, as they belong to reactions with same olefin type, solvent, ligand, and so on (Fig. 7a). Consider a practically likely situation wherein one wants to predict the selectivity of a few AHO reactions, in the initial phase of reaction development. These reactions will form the test set. Now, if we use the standard meta-learning framework, a part of this test data will be needed as support set to obtain predictions on the query set (Fig. 3c). Since, these are new reactions for which the experimental data is not available, additional wet-lab experiments will be required to obtain the support set data at meta-test time.

However, the second approach (meta-cluster) samples the support set from the training data that is similar to the query set at meta-test time (Fig. 5b). As the support set of the test task is not used in the meta-cluster approach, no new experiments are required to obtain predictions on the query set at meta-test time. Following the process described in Fig. 5a, training and test tasks are constructed. The meta-trained model and the test data remains same in both standard and clustering-based approaches. It is clear from Fig. 7b that Protonet_cluster with an AUPRC score of $0.9147 \pm 0.0022$ performs better than RF_cluster and GNN_cluster which results in an AUPRC score of $0.8692 \pm 0.0017$ and $0.8021 \pm 0.0071$, respectively. Also, Protonet and Protonet_cluster achieves comparable performance with all support

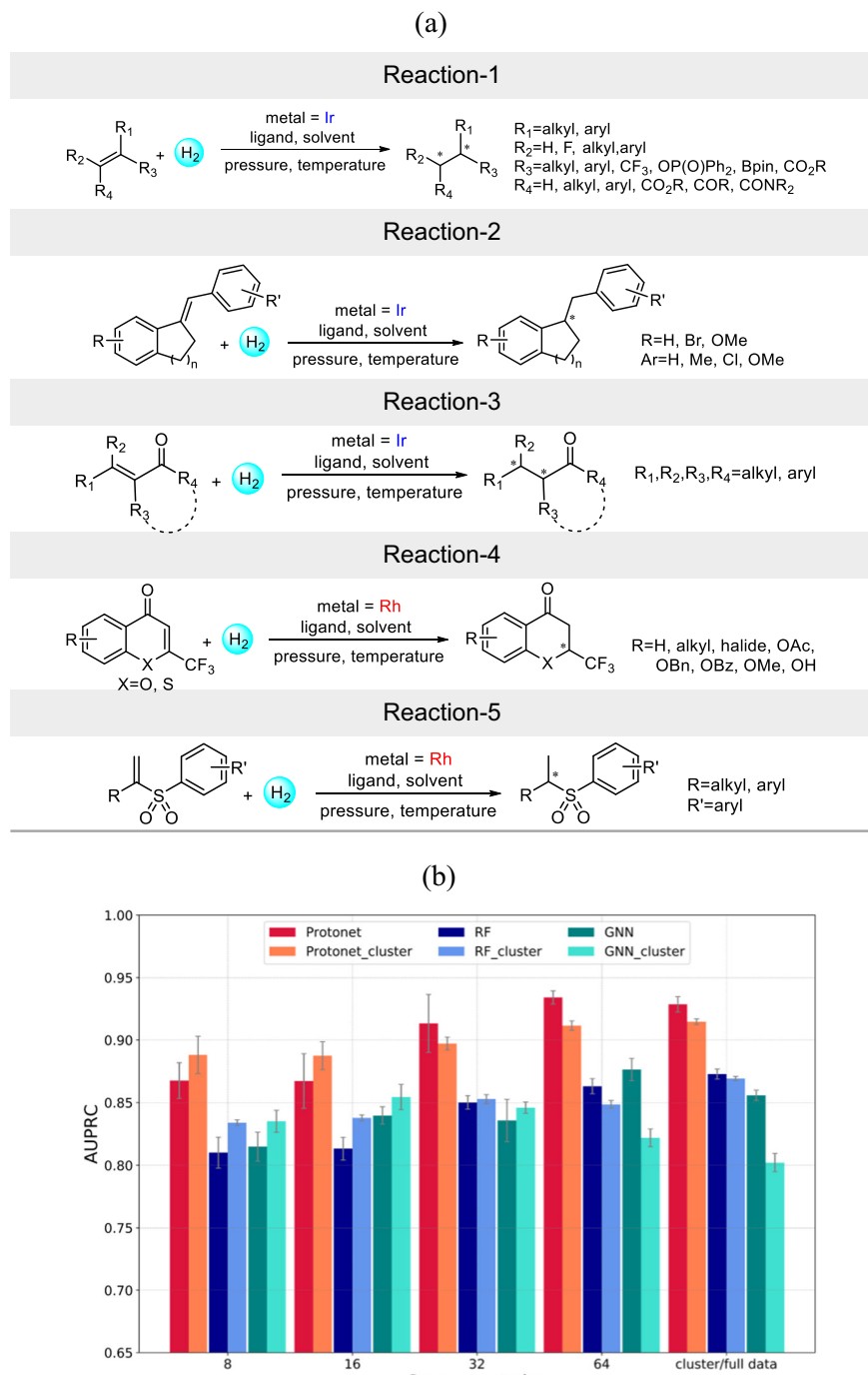

**Fig. 7 | Meta-model performance on out-of-sample test set. a** A general scheme of reactions considered in out-of-sample test set. **b** Comparison of performance of different meta-learning and single-task methods on the out-of-sample test set. The best performance is obtained from the meta-learning method (shown as Protonet and Protonet_cluster).

set sizes except 64, where Protonet performs slightly better than Protonet_cluster. Thus, the meta-cluster approach can provide good initial estimates of the reaction performance. This is particularly useful in scenarios where performing experiments are time- and resource-intensive. These results further suggest the general applicability of our meta-learning approach to enhance the predictions on the literature datasets.

In conclusion, we have demonstrated a proof-of-concept application of meta-learning in asymmetric catalysis to identify reactions with high enantioselectivity. Although this approach can find broader applicability in reaction development, we have chosen asymmetric hydrogenation of olefins to convey the potential of our meta-learning model. The model is meta-trained to capture the knowledge shared among diverse training tasks such that it can efficiently adapt to the unseen test data. Prototypical network is selected as the choice of meta-learning model. A comparison with popular machine learning methods such as random forests and graph neural networks is also done. We show the utility of our meta-trained model to make predictions with only a limited amount of data in the training set. Prototypical networks are found to return better predictions as compared to RF and

GNN in every data setting. Prototypical networks achieved an AUPRC score of $0.9117 \pm 0.0026$ with a mere 64 training reactions. It is significantly greater than RF and GNN that yielded an AUPRC score of only $0.8369 \pm 0.0055$ and $0.8259 \pm 0.0021$ respectively, even with full training data. The generalizability of the pretrained meta-model is further evaluated on an out-of-sample test set. An alternative approach

(a)

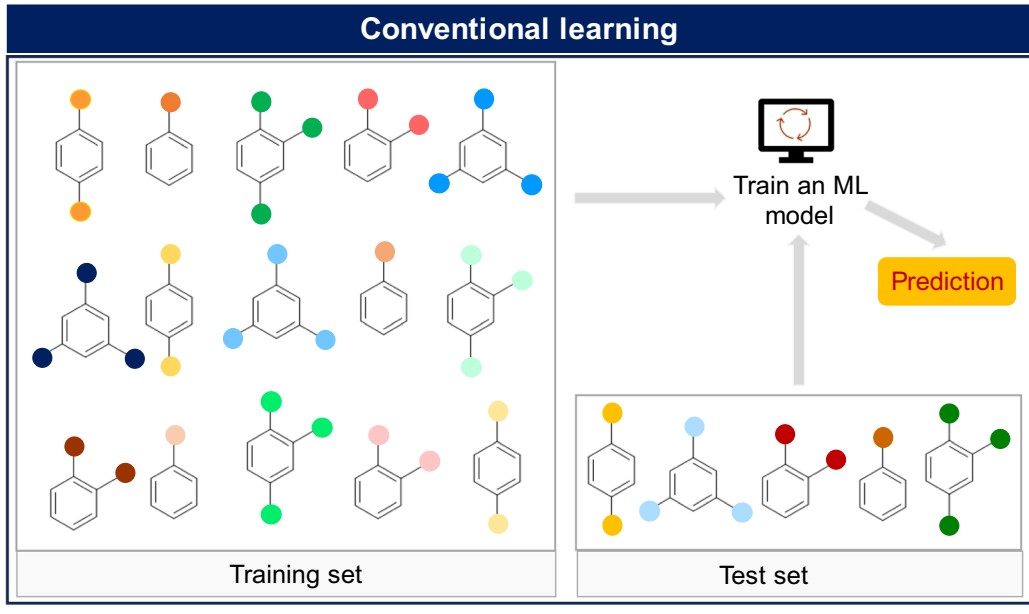

(b)

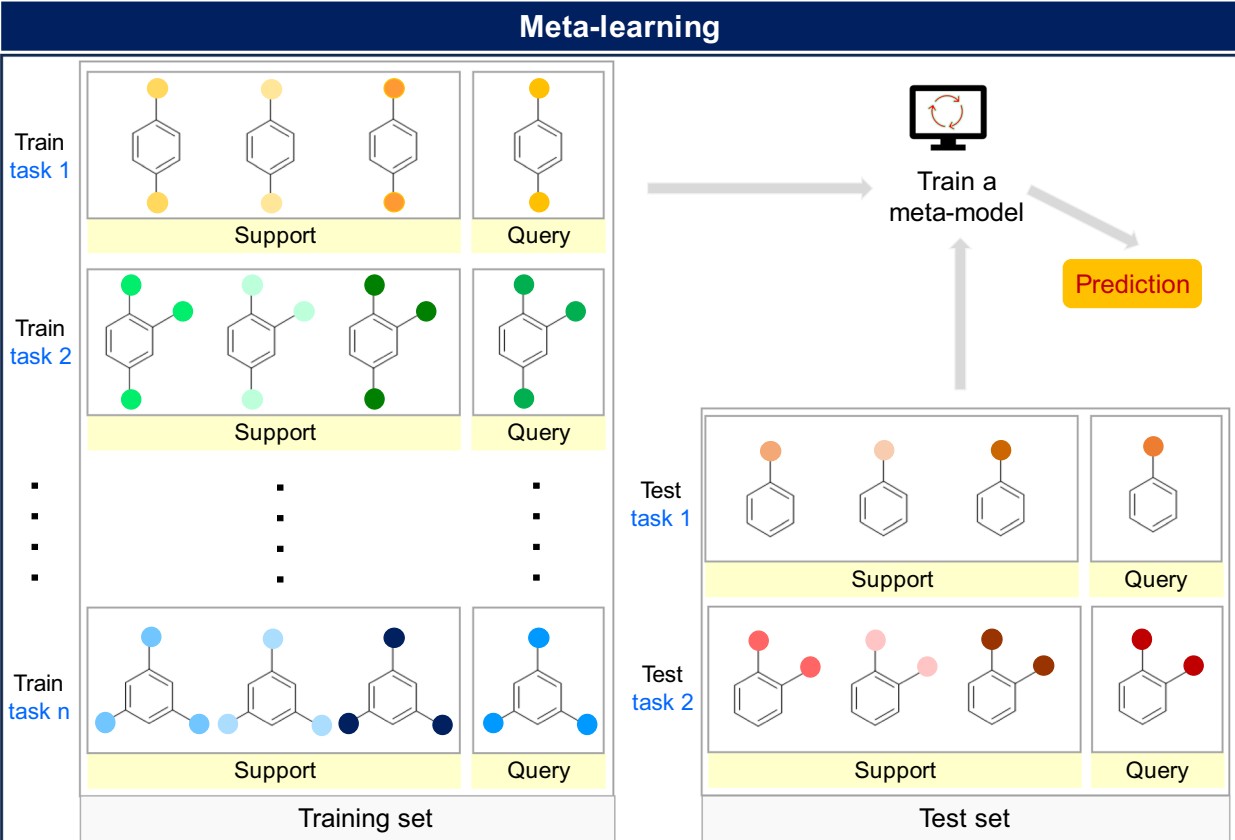

**Fig. 8 | Overview of conventional learning and meta-learning approaches.** Illustration of (**a**) conventional learning and (**b**) meta-learning approaches. A single large training dataset is required in conventional learning to train an ML model to obtain predictions on the test set. Meta-learning utilizes the information extracted from a set of related tasks to generalize and make predictions on unseen test tasks.

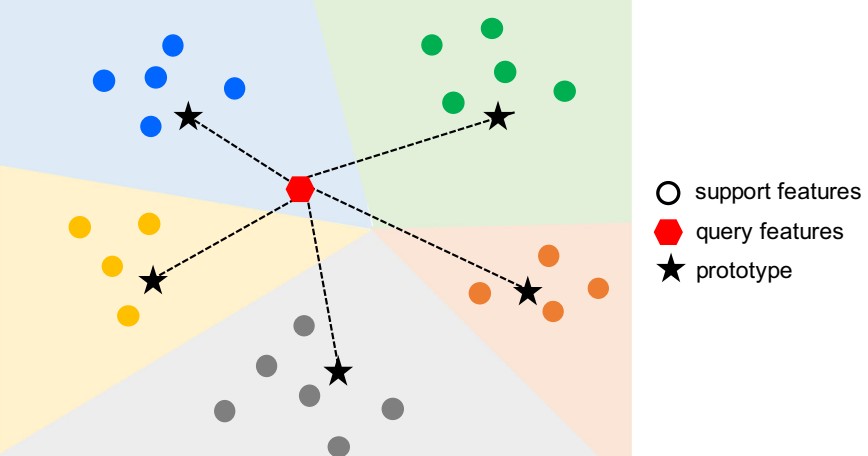

**Fig. 9 | Prototypical networks.** An overview of how the prototypical network meta-learning methods employed in this study make predictions. For each class, a *prototype* is first computed as the mean vector of the support set embeddings. The query set embeddings are then classified by calculating their Euclidean distance with each prototype.

for meta-test is proposed where the support set for test tasks is sampled from the training data. This method is found to provide good predictive performance on the out-of-sample test set. It can be useful in the early stages of reaction development to get reasonable estimates of the reaction performance, without needing to perform new experiments. Thus, our proposed approach is an important addition to the standard meta-learning framework. These findings suggest that our meta-learning protocol can be tailored as an effective tool to exploit the vast amounts of literature data for building a prediction model for any reaction of interest. Our meta-model would be of great value when integrated into the reaction discovery pipeline, as it can work well with a relatively small number of reactions.

## Methods

Standard supervised algorithms can be viewed as single-task methods that rely on one dataset $\mathscr{D}$ to train a ML model (Fig. 8a). The model performance is then evaluated by making predictions on a held-out test set. These approaches in general require large amounts of training examples following a distribution similar to that of the test set. Meta-learning algorithms, on the other hand, require a set of related supervised tasks $\mathscr{D}_{train} = \{\mathscr{T}_t\}_{t=1}^T$ for training (Fig. 8b). Each task $\mathscr{T}_t = \{(x_i, y_i)\}_{i=1}^{N_{\mathscr{T}}}$ is further divided into a support set $\mathscr{T}_{t,support}$ and a query set $\mathscr{T}_{t,query}$. During meta-training, the model is trained to make accurate predictions on the query set, given the support set examples. At meta-testing time, the unseen tasks $\mathscr{T}_u \in \mathscr{D}_{test}$ are used to evaluate the performance of the meta-trained model by measuring prediction error on $\mathscr{T}_{u,query}$, with access to labels and features of $\mathscr{T}_{u,support}$ (Fig. 8b).

In this work, we have considered a popular meta-learning approach, namely, prototypical networks (Fig. 9). The prototypical network[54] is a metric-based meta-learning approach that focuses on learning a distance metric to measure the similarity between data samples. An embedding function such as a neural network is used to generate the support and query embeddings. Given these embeddings, a prototype is computed first as the mean vector of the support set embeddings for each class. The query set embeddings are then classified by calculating their distance with each prototype. The Euclidean distance is usually a popular choice for the distance metric. Subsequently, a distribution over classes for a query sample is obtained by applying softmax over negative distances to prototypes. The model is trained by minimizing the negative log-likelihood across training task query sets. The prototypical networks are primarily suitable for classification problems.

## Data availability

The data is available through http://asymcatml.net and GitHub (https://github.com/sukriti243/Meta-learning-for-selectivity-prediction). Additional figures, tables and technical details are provided in the Supplementary Information. Data supporting the findings of this manuscript are also available from the corresponding author upon request.

## Code availability

The code is publicly available through GitHub (https://github.com/sukriti243/Meta-learning-for-selectivity-prediction).

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

## Acknowledgements
J.M.H. and S.S. acknowledge support from a Turing AI Fellowship under grant EP/V023756/1.

## Author contributions
S.S.: conceptualization, investigation, methodology, analysis, writing the original draft. J.M.H.: analysis, writing-review, editing, supervision.

## Competing interests
The authors declare no competing interests.
