## [Transparent Peer Review file · Nature Communications]

A Meta-learning Approach for Selectivity Prediction in Asymmetric Catalysis

Corresponding Author: Dr Sukriti Singh

Version 0:

Reviewer comments:

Reviewer #1

(Remarks to the Author)

The manuscript introduces an interesting approach in the use of meta-learning to enhance predictability in asymmetric catalysis, focusing particularly on the asymmetric hydrogenation of olefins. By utilizing a prototypical network as part of a meta-learning framework, the authors propose a useful method to extrapolate from limited data to predict reaction outcomes effectively. This method has demonstrated superiority over traditional machine learning techniques, which highlights a significant advancement in leveraging minimal data for structure-performance relationship predictions, especially for a challenging task like AHO. This study provides an insightful exploration into meta-learning's potential within organic chemistry, it would be an excellent addition to Nature Communications if the authors address the following minor concerns to clarify the robustness and applicability of their findings.

Minor Issues:

1. Given the distinct mechanistic implications of metal and ligand variations in asymmetric hydrogenation, which can lead to different hydrogenation pathways (inner- vs. outer-sphere mechanisms), it's crucial to understand how the proposed query-based method handles these variations. Does the model implicitly differentiate between these mechanistic domains, or does it treat them uniformly? An analysis from this perspective would be enlightening. If the model does not inherently separate these domains, implementing a data splitting strategy based on mechanistic insights before querying could potentially refine the predictions and enhance model performance.
2. The method relies on querying within a structure-performance space, a process heavily dependent on the way molecules are projected into this space. While molecular fingerprints are a common method for such projections, a comprehensive survey of alternative techniques could provide broader insights and possibly reveal better methods for establishing these correlations. This could lead to improved predictive accuracy and broader applicability of the model.
3. Enhancing a model through mechanistically informed data augmentation is promising; however, such approaches can sometimes face challenges when applied to test sets with narrow value ranges (e.g., 0-50% ee) that do not resemble the training domain's data distribution. It would be beneficial for the authors to construct challenging test sets with varied data ranges to examine if their model can maintain effectiveness across different value domains. This analysis could help in assessing the model's generalization capabilities and its utility in practical scenarios where data diversity is limited.

(Remarks on code availability)

Reviewer #2

(Remarks to the Author)

In this manuscript, the authors studied "meta-learning" methods on reaction outcomes predictions. By training a neural network, the authors can map reaction features (fingerprints or GNN encoded) into a latent space, where the authors can measure the distance between support set and query reaction during inference to determine if the reaction is high selective or low selective. The authors compare the developed model to conventional random forest and GNN on multiple training/testing splitting strategies and showed the developed model outperforms others. This ML strategy that optimizes the

parameter by minimizing the distance between query and support should be interest of broad comp chem/cheminformatics researchers, while the manuscript needs additional experiments/data to be published.

1. The authors need to publish the dataset used in this work. I appreciate the authors publish the code used in this work, but for the purpose of reproduction, the dataset should also be provided. In the code, the training/testing data was read from a local file, which is not included in the current package.
2. In the third benchmarking task (out-of-sample test set), Protonet is only slightly better than other methods, which is not as outperformed in the previous two benchmarking tasks. Could the authors comment on this? For the first two benchmarking tasks, Protonet performance might be biased by the single random split between training and testing set. Could the authors provide results on the first two benchmarking tasks through cross-validation? For example, split training/testing in 5 or 10 folds so that each reaction will at least be tested once in the testing set.
3. I'm not sure I saw how are parameters in the GNN for the reaction encoding (fig 4b) determined?
4. I wonder does the ProtoNet model bring reaction features to a latent space where reactions with similar outcomes have a closer distance? Could the authors add a baseline into benchmarking, in which the neural network is skipped, but only apply the distance metric to the reaction features to classify the query reactions. This baseline method should not require any new data and is fully based on distance between encoded reaction features, and could give us insight on the role of the NN in ProtoNet.
5. The meta-learning should be trained on different tasks from the testing task, in which testing cases should have no close-ins in training case. In the present work, although authors split training and testing into many tasks either randomly or through clustering, the testing reaction still have close-ins in the training set. I don't feel this is meta-learning. The authors should consider train and test on more different tasks, for example, splitting the training/testing by metals, ligand, solvent, pressure, or simply leave one cluster in Fig 7 out as the testing set.

Some minor comments:

1. Should the circle in legend of Fig2 uncolored and referring to all circles in all classes, instead of being blue and referring one class?
2. How are number of tasks in train/test determined?

(Remarks on code availability)

Please see comments for authors point 1

Version 1:

Reviewer comments:

Reviewer #1

(Remarks to the Author)

My previous concerns have been addressed by the authors. I support its publication as is.

(Remarks on code availability)

Reviewer #2

(Remarks to the Author)

I appreciate the authors' responses. The response well addressed some of my concerns, however for other concerns I feel the authors could provide a better solution.

R2-C3 : My concern was more about the way the authors divided train-valid-test. Although the authors split train-valid-test randomly, the single splitting may result in a biased testing set, which could bias the performance of the model on testing. A safer way to assess the model on this dataset is to have the model evaluated on each datapoint. To achieve that, the data will need to be split, e.g. 5 times, depending on the size of training and testing. And the model will be exposed to evaluation on each datapoint once. A distribution of the performance on the 5 training/testing can be reported. I appreciate the authors reported the cross-validation on the 10 fold splitting of the support-query in the testing data, but there can still be bias from the splitting of training/testing. Such multi-fold splitting between training and testing is widely used for reaction outcome predictions (e.g. doi: 10.1039/D0SC04823B). I would suggest the authors also consider such evaluation for their model.

R2-C6: I appreciate the authors added the clustering split into SI. I think this result worth to be added into the main body of article and need more details and discussions. For the result in S11, more details are needed, for example, which clusters are used as testing and size of each cluster, et al. For the numbers in S11, it's surprisingly to see the model is performing even better than splitting the same cluster into training/testing as in Figure 7. Having different clusters in testing should be a more challenging task than having data from the same cluster in training and testing. However, the result showed in S11 and Figure7 are counterintuitive. I think a cross-validation of which cluster is used as testing could help. Could the authors please add discussions on this in the main body of article, not just a table with no detailed methods in SI. I think this is important and is also aligned with another reviewer's comment R1-C1 and R1-C4. Ideally, I would also like to see a testing/training splitting based on mechanism as suggested by another reviewer, which is what meta-learning should be used for, while the authors argued this is challenging to do and outside of the scope of the present study. What I suggested

here as to split training/testing by clusters serves the same goal and should be a doable alternative, and I hope the authors can provide a more satisfying answer.

(Remarks on code availability)

Data is readable on the website, but not sure if readers could download and reuse the dataset to reproduce the work.

Version 2:

Reviewer comments:

Reviewer #2

(Remarks to the Author)

My concerns are well addressed in this revision, the manuscript is suitable for publication.

(Remarks on code availability)

Reviewer 1:

R1-C1 The manuscript introduces an interesting approach in the use of meta-learning to enhance predictability in asymmetric catalysis, focusing particularly on the asymmetric hydrogenation of olefins. By utilizing a prototypical network as part of a meta-learning framework, the authors propose a useful method to extrapolate from limited data to predict reaction outcomes effectively. This method has demonstrated superiority over traditional machine learning techniques, which highlights a significant advancement in leveraging minimal data for structure-performance relationship predictions, especially for a challenging task like AHO. This study provides an insightful exploration into meta-learning's potential within organic chemistry, it would be an excellent addition to Nature Communications if the authors address the following minor concerns to clarify the robustness and applicability of their findings.

We thank the reviewer for appreciating our work.

R1-C2 Minor Issues:

1. Given the distinct mechanistic implications of metal and ligand variations in asymmetric hydrogenation, which can lead to different hydrogenation pathways (inner- vs. outer-sphere mechanisms), it's crucial to understand how the proposed query-based method handles these variations. Does the model implicitly differentiate between these mechanistic domains, or does it treat them uniformly? An analysis from this perspective would be enlightening. If the model does not inherently separate these domains, implementing a data splitting strategy based on mechanistic insights before querying could potentially refine the predictions and enhance model performance.

We thank the reviewer for this insightful suggestion for constructing tasks for meta-learning. The dataset used in this study comprises of over 12,000 AHO reactions catalyzed by Ir, Rh, and Co metal catalysts. There are thousands of distinct olefins and ligands. The mechanism generally varies with the type of metal, olefin, and ligand. Given the diversity and large number of reactions, it is extremely difficult to decipher the mechanism of each reaction. Therefore, a data splitting strategy based on mechanistic insights is out of scope of the present work. However, we wish to explore this alternative aspect of constructing tasks for meta-learning in our future work.

R1-C3 2. The method relies on querying within a structure-performance space, a process heavily dependent on the way molecules are projected into this space. While molecular fingerprints are a common method for such projections, a comprehensive survey of alternative

techniques could provide broader insights and possibly reveal better methods for establishing these correlations. This could lead to improved predictive accuracy and broader applicability of the model.

The two main approaches to represent molecules are using non-learned and learned representations. In this work, we have considered both featurization methods to represent the reaction. The first is molecular fingerprint which is a non-learned representation. The other representation utilizes molecular graphs which is a learned representation. Graph neural networks are popular for various molecular property prediction and reaction outcome prediction tasks. Here, we use a message-passing neural network to obtain graph embeddings of the reaction components. The details are given in the section “Feature representation” in the manuscript (Figure 4). The implementation details of GNNs are provided in section 1.1 of the Supplementary Information. The model performance of both feature representations is found to be comparable (Table S1 and S3 in the Supporting Information).

RI-C4 3. Enhancing a model through mechanistically informed data augmentation is promising; however, such approaches can sometimes face challenges when applied to test sets with narrow value ranges (e.g., 0-50% ee) that do not resemble the training domain’s data distribution. It would be beneficial for the authors to construct challenging test sets with varied data ranges to examine if their model can maintain effectiveness across different value domains. This analysis could help in assessing the model's generalization capabilities and its utility in practical scenarios where data diversity is limited.

We have taken following measures to show that our results are positive given the limitation of the existing data:

- (1) We have used the area under the precision-recall curve (AUPRC) to evaluate the classification performance. The AUPRC score is sensitive to class imbalance. The results of our study indicate that the meta-learning models return reasonable predictions even with the imbalanced data.
- (2) We have also reported the confusion matrices for model performance which suggest that the model is able to provide good predictions even in low %ee range (Table S9 in the Supplementary Information).
- (3) We have also evaluated the performance of our approach with different classification thresholds. Two classification thresholds are used: 70 and 90. We obtained good AUPRC scores, suggesting good predictive performance in low %ee range (section 6 in the Supplementary Information).

(4) The generalizability of our meta-model is demonstrated on an out-of-sample test set that is not a part of the original training data. The meta-learning method is found to provide good predictive performance on the out-of-sample test set in terms of AUPRC score.

Reviewer 2:

R2-C1 In this manuscript, the authors studied “meta-learning” methods on reaction outcomes predictions. By training a neural network, the authors can map reaction features (fingerprints or GNN encoded) into a latent space, where the authors can measure the distance between support set and query reaction during inference to determine if the reaction is high selective or low selective. The authors compare the developed model to conventional random forest and GNN on multiple training/testing splitting strategies and showed the developed model outperforms others. This ML strategy that optimizes the parameter by minimizing the distance between query and support should be interest of broad comp chem/cheminformatics researchers, while the manuscript needs additional experiments/data to be published.

We thank the reviewer for appreciating our work. We have now included all the changes as suggested by the reviewer. We hope that with these modifications the reviewer would find our manuscript suitable for publication.

R2-C2 1. The authors need to publish the dataset used in this work. I appreciate the authors publish the code used in this work, but for the purpose of reproduction, the dataset should also be provided. In the code, the training/testing data was read from a local file, which is not included in the current package.

As provided in reference 39 in the manuscript, the full dataset is available at <http://asymcatml.net>.

R2-C3 2. In the third benchmarking task (out-of-sample test set), Protonet is only slightly better than other methods, which is not as outperformed in the previous two benchmarking tasks. Could the authors comment on this? For the first two benchmarking tasks, Protonet performance might be biased by the single random split between training and testing set. Could the authors provide results on the first two benchmarking tasks through cross-validation? For example, split training/testing in 5 or 10 folds so that each reaction will at least be tested once in the testing set.

A better performance is observed when the test set has more similarity to the training set. In Figure 6a, the training and test sets are randomly sampled and therefore provides the lowest AUPRC score. In Figure 7c, the training and test are relatively similar as they are formed

using the clustering approach. Therefore, a performance improvement is noted when compared to Figure 6a. The plot in Figure 8b corresponds to the out-of-sample test set performance. The training and test sets formed from the out-of-sample test data are very similar, as they belong to reactions with same olefin type, solvent, ligand, and so on. This results into the model performance better than that in Figure 7c. Also, we wish to clarify that we have reported the model performance averaged over ten different support-query random splits of every test task.

R2-C4 3. I'm not sure I saw how are parameters in the GNN for the reaction encoding (fig 4b) determined?

The architecture detail of GNN is provided under the section “Feature representation” (page 7). The details on training GNN hyperparameters is provided in section 1.1 in the Supplementary Information. It is also briefly described here.

The molecular graphs of olefin, ligand, and solvent are fed into graph neural networks (GNNs) to extract feature embeddings. For each reaction component, the graph with atom and bond features is passed to the message-passing neural network. Three message-passing steps are considered to obtain the node representation, where an edge network is used as the message function and a gated recurrent unit (GRU) as the update function. The node representation vector has a dimension of 64. In the readout step, a set2set model (with number of set2set layers fixed to 3) provides global pooling over the node representation vectors. This results into a graph representation vector of dimension 512. In this case, the full reaction representation is a concatenation of the graph embeddings with one-hot encoded vectors, and reaction conditions (Fig. 4b). The reaction representation vector thus obtained has a dimension of 1544.

The 1544-dimensional graph-based reaction representation are used with GNN (Figure 4b). This is passed through two fully-connected layers with dimensions 1024 and 512, followed by an output layer. A dropout rate of 0.1 is applied to these layers with PReLU activation function. The model is then trained for 400 epochs using the Adam optimizer with a learning rate of 0.001 and a batch size of 512.

R2-C5 4. I wonder does the ProtoNet model bring reaction features to a latent space where reactions with similar outcomes have a closer distance? Could the authors add a baseline into benchmarking, in which the neural network is skipped, but only apply the distance metric to the reaction features to classify the query reactions. This baseline method should not require any new data and is fully based on distance between encoded reaction features, and could give us insight on the role of the NN in ProtoNet.

As suggested by the reviewer, we removed the neural network and evaluated the performance of prototypical networks (Table S10 in the revised Supplementary Information).

It is noted that the performance of prototypical networks without the embedding function is considerably lower as compared to with neural network (Table S8 in the Supplementary Information).

In prototypical networks, neural networks are used as an embedding function with learnable parameters. If neural network is removed from prototypical networks, the model will have no parameters to learn and update the embeddings of support set.

R2-C6 5. The meta-learning should be trained on different tasks from the testing task, in which testing cases should have no close-ins in training case. In the present work, although authors split training and testing into many tasks either randomly or through clustering, the testing reaction still have close-ins in the training set. I don't feel this is meta-learning. The authors should consider train and test on more different tasks, for example, splitting the training/testing by metals, ligand, solvent, pressure, or simply leave one cluster in Fig 7 out as the testing set.

As suggested by the reviewer, we constructed the meta-learning tasks by splitting based on clusters. Out of 10 clusters, we kept 8 for training and 2 for test. The results are provided in Table S11 in the revised Supplementary Information. Our meta-learning model obtained a good predictive performance in this setting of train-test tasks as well.

R2-C7 Some minor comments:

1. Should the circle in legend of Fig2 uncolored and referring to all circles in all classes, instead of being blue and referring one class?

We have made changes to Figure 2 in the revised manuscript as suggested by the reviewer.

R2-C8 2. How are number of tasks in train/test determined?

The detail of task construction for meta-learning is provided in sections 1.2 and 4 in the Supplementary Information.

Reviewer 2:

R2-C1 I appreciate the authors' responses. The response well addressed some of my concerns, however for other concerns I feel the authors could provide a better solution.

We thank the reviewer for appreciating our response. We have now included additional experiments as suggested by the reviewer in the revised manuscript and supplementary information.

R2-C2 R2-C3: My concern was more about the way the authors divided train-valid-test. Although the authors split train-valid-test randomly, the single splitting may result in a biased testing set, which could bias the performance of the model on testing. A safer way to assess the model on this dataset is to have the model evaluated on each datapoint. To achieve that, the data will need to be split, e.g. 5 times, depending on the size of training and testing. And the model will be exposed to evaluation on each datapoint once. A distribution of the performance on the 5 training/testing can be reported. I appreciate the authors reported the cross-validation on the 10 fold splitting of the support-query in the testing data, but there can still be bias from the splitting of training/testing. Such multi-fold splitting between training and testing is widely used for reaction outcome predictions (e.g. doi: 10.1039/D0SC04823B). I would suggest the authors also consider such evaluation for their model.

As per reviewer's suggestion, we have also reported the performance of meta-learning and single-task methods on five different train-test splits (Tables S1, S2, S4, S8, and S12).

R2-C3 R2-C6: I appreciate the authors added the clustering split into SI. I think this result worth to be added into the main body of article and need more details and discussions. For the result in S11, more details are needed, for example, which clusters are used as testing and size of each cluster, et al. For the numbers in S11, it's surprisingly to see the model is performing even better than splitting the same cluster into training/testing as in Figure 7. Having different clusters in testing should be a more challenging task than having data from the same cluster in training and testing. However, the result showed in S11 and Figure7 are counterintuitive. I think a cross-validation of which cluster is used as testing could help. Could the authors please add discussions on this in the main body of article, not just a table with no detailed methods in SI. I think this is important and is also aligned with another reviewer's comment R1-C1 and R1-C4. Ideally, I would also like to see a testing/training splitting based on mechanism as suggested by another reviewer, which is what meta-learning should be used for, while the authors argued this is challenging to do and outside of the scope of the present study. What I suggested here

as to split training/testing by clusters serves the same goal and should be a doable alternative, and I hope the authors can provide a more satisfying answer.

We apologize for not being clear. We have added new experiments in the revised manuscript and SI as per reviewer's suggestions. We evaluated the performance of meta-learning method on more challenging cluster-based splits. A leave-one-cluster-out (LOCO) approach where one cluster is kept as test task, while remaining clusters are used as training tasks is considered. This results in a combination of 10 different train-test tasks. We have included additional text and figure in the revised manuscript (page 15, Fig. 8). The model performance averaged over ten different support-query random splits of every test task is reported in Table S16. A performance comparison with single-task methods trained on full training data is provided in Table S17. The results indicate that meta-learning is able to generalize better than single-task methods across various clusters.

R2-C4 Data is readable on the website, but not sure if readers could download and reuse the dataset to reproduce the work.

The dataset can be obtained after registering on the website.